# AdaTok: Adaptive Visual Representation with Quality-Preserving Dynamic Tokens

## Abstract

Image tokenization, a cornerstone of modern visual representation, faces a fundamental dilemma posed by content diversity. A fixed number of tokens is inherently suboptimal, causing computational redundancy for simple images and risking information loss for complex ones. While variable-length methods offer a potential solution, they are typically empirical and heuristic, lacking a theoretical mechanism for adaptation. To address this dilemma, we propose **AdaTok**, an adaptive visual representation framework with high flexibility for diverse representational needs. Specifically, it incorporates an elastic encoder capable of encoding an image into an arbitrary number of tokens. Building on this, we design a novel token selection strategy: guided by the information bottleneck principle, it enables the model to learn a policy that maximizes representational information under a minimal budget. This allows AdaTok to autonomously find a sufficient yet compact token set for each image. Extensive experiments demonstrate that this elastic, sample-level tokenization yields superior performance in both image reconstruction and generation. By preserving essential details while minimizing redundancy, AdaTok not only enhances efficiency but also creates a more natural alignment with the variable-length structure of language, paving the way for more unified and efficient vision-language models (VLMs). Code is available at *anonymous.4open.science/r/AdaTok*.

## 1 Introduction

Image tokenization is crucial in visual tasks as it both compresses data representations and facilitates their adaptation for downstream tasks, which can be divided into continuous and discrete tokenizers. Continuous tokenizers such as VAE (Kingma et al., 2013) are primarily utilized in the denoising generation process of diffusion models (Dhariwal & Nichol, 2021; Rombach et al., 2022; Hoogeboom et al., 2023), whereas discrete tokenizers like VQVAE (Van Den Oord et al., 2017) and FSQ (Mentzer et al., 2023) are employed in the autoregressive (AR) image generation process (Esser et al., 2021; Yu et al., 2023a;b; Ma et al., 2024; Peebles & Xie, 2023). These tokenizers are all 2D-based, maintaining a spatial correspondence (i.e., *the top-left latent token aligns with the top-left image patch*). This limits the flexibility of visual representations, as each token corresponds to the compressed representation of a single image patch, losing global context. Additionally, it constrains all images to be represented using a fixed number of tokens.

Recognizing this rigidity, recent works have begun exploring more flexible representations. TiTok (Yu et al., 2025) breaks the structural limitations of conventional tokenization for the first time by representing images as a one-dimensional sequence, thereby achieving a more compact visual representation. Methods like FlexTok (Bachmann et al., 2025) and One-D-Piece (Miwa et al., 2025) demonstrated that a tokenizer could be trained to reconstruct images from a variable number of tokens. However, these methods only provide the capability for flexible-length encoding; they lack a mechanism to autonomously determine the appropriate number of tokens for a given image. Other approaches have tried to solve this decision problem, but with practical drawbacks. Semanticist (Wen et al., 2025) imposes a principled structure, such as a PCA-like mechanism to ensure tokens are ordered by informational importance, but still do not decide the optimal token count. Others tackle the decision directly: ElasticTok (Yan et al., 2024) employs a costly search process; another contemporary work (Duggal et al., 2024) uses a recursive, iterative refinement to grow the token set; and the Content-Adaptive Tokenizer (CAT) (Shen et al., 2025) leverages an external Large Language Model (LLM) to

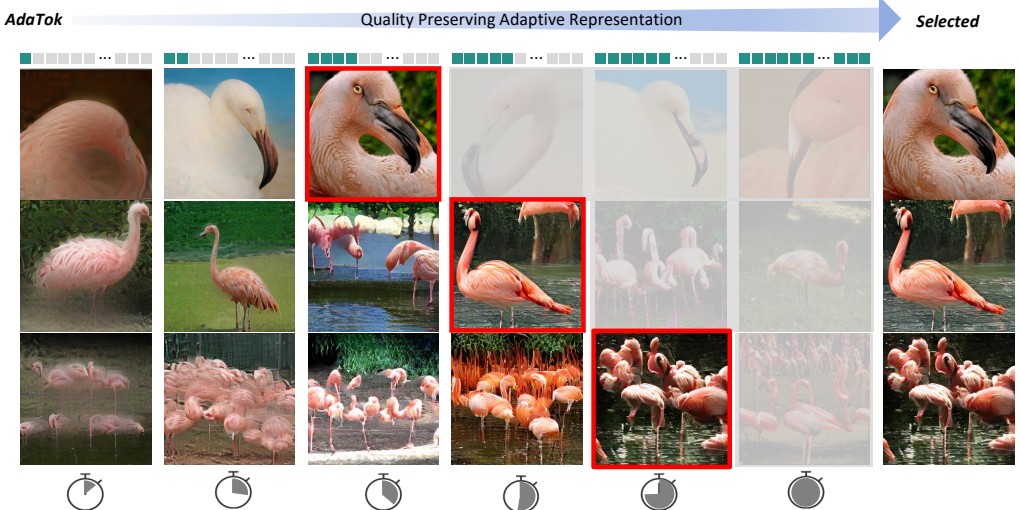

Figure 1: **Motivation of AdaTok.** The complexity of images varies, both in encoding and generation. Similar to sentences, images should be represented at an adequate quality, without unnecessary continuation of the representation.

predict complexity from captions. While innovative, these methods either rely on computationally intensive search, or depend on external models and textual data, making the decision process indirect and inefficient for inference.

The need for adaptive tokenization is magnified in modern Vision-Language Models (VLMs), which suffer from a structural mismatch between variable-length text and fixed-length visual inputs. This rigidity causes both computational inefficiency and a bottleneck for fine-grained reasoning. To address this, we must first answer the question: *1) Should different images be represented by the same number of tokens?* Intuitively, and by analogy to language where length correlates with content (Gibson, 1998; Lin, 2004; Piantadosi et al., 2011), the answer is no; complex images require more tokens than simple ones (Bachmann et al., 2025; Miwa et al., 2025; Yan et al., 2024). This immediately raises the pivotal second question: *2) How can we decide the appropriate number of tokens for an image?* We posit that the principle of efficient communication in language—conveying essential meaning—offers a guide. An image representation could be guided by the information bottleneck principle, retaining just enough information to preserve key features for reconstruction. Thus, reconstruction quality itself can signal when the token set is sufficient, marking a natural stopping point for token allocation.

In this paper, we introduce **AdaTok**, a novel framework that materializes this vision by learning to represent images with an adaptive, content-aware number of tokens. AdaTok consists of two core components: a flexible encoder trained for variable-length robustness, and an autonomous token selection strategy grounded in information bottleneck (IB) theory. *To achieve representational flexibility*, we build upon the 1D tokenizer paradigm and introduce a nested masking strategy during training. This forces the model to prioritize information, ordering tokens by importance and enabling high-quality reconstruction from varied token lengths. *To enable autonomous token selection*, we design a lightweight policy network that efficiently predicts the optimal token count in a single forward pass, directly from image features. This policy is trained to balance the trade-off between representational compactness (fewer tokens) and reconstruction fidelity, effectively learning to identify the "information bottleneck" for each input. AdaTok's principled adaptability enables more efficient and context-aware visual processing. Demonstrating enhanced flexibility across image reconstruction, generation, and task transfer, our model creates a more natural alignment with the variable-length structure of language, paving the way for more unified vision-language models (VLMs). Moreover, it delivers substantial gains in compression and generation speed while maintaining quality, highlighting its significant practical potential. Our contributions are:

- We propose AdaTok, a novel framework that empowers a visual tokenizer to autonomously select a content-aware number of tokens for each image.

- We introduce a principled approach for adaptive tokenization by formulating the token selection process as a learnable policy guided by the Information Bottleneck theory.
- We demonstrate that AdaTok's representational flexibility not only yields significant gains in efficiency and downstream task performance, but also fosters a more natural alignment between vision and language modalities by embracing variable-length structures.

## 2 RELATED WORKS

Compressing high-resolution images into a compact latent space is a foundational technique in computer vision (Ronneberger et al., 2015; Hinton & Salakhutdinov, 2006). An early and influential approach is continuous tokenization via Variational Autoencoders (VAEs) (Kingma et al., 2013), which encode images into a continuous latent space. This paradigm is fundamental to modern generative models like Stable Diffusion (Rombach et al., 2022), which operates its entire denoising process within this compressed space to achieve high-fidelity synthesis. In a parallel effort to bridge visual representation with sequence-based models, discrete tokenization was pioneered by VQ-VAE (Van Den Oord et al., 2017). This method uses a learnable codebook to quantize the latent space into a grid of discrete tokens. To improve the perceptual quality of these discrete representations, VQGAN (Esser et al., 2021) integrated adversarial training from GANs (Goodfellow et al., 2014), establishing the 2D discrete tokenization as a cornerstone for many subsequent autoregressive models.

To overcome the rigidity of fixed-grid 2D tokenizers, a recent line of work has shifted towards structure-free, 1D representations. TiTok (Yu et al., 2025) pioneered this direction by demonstrating that an image could be effectively represented as a single, compact one-dimensional sequence, breaking the conventional 2D structural constraints. Building upon this concept, subsequent methods explored the idea of flexible-length tokenization. For instance, FlexTok (Bachmann et al., 2025) and One-D-Piece (Miwa et al., 2025) developed tokenizers capable of reconstructing images from a variable number of tokens, proving the feasibility of adaptive encoding. However, these models provided the capability without an autonomous mechanism to decide the token count. Addressing this challenge, ElasticTok (Yan et al., 2024) introduced a method to directly determine the optimal number of tokens for a given image, albeit through a computationally intensive search process.

Once visual tokens are obtained, their generation has evolved significantly. While traditional autoregressive (AR) models are slow due to their sequential nature, this bottleneck has been addressed by parallel decoding schemes like the mask-and-predict method in MaskGIT (Chang et al., 2022). Beyond speed, recent innovations have also introduced new paradigms, including the "next-scale" coarse-to-fine strategy of VAR (Tian et al., 2025), autoregressive modeling in continuous latent space (MAR (Li et al., 2024)), and the direct adaptation of large language model (LLM) architectures for unified generation (LlamaGen (Sun et al., 2024)). While existing methods are limited by either fixed-length representations that clash with language models or by non-adaptive variable-length schemes, we introduce AdaTok, which gives images a language-like variable-length property, paving the way for the development of truly unified Vision-Language Models (VLMs).

## 3 PRELIMINARIES

In this section, we present the foundational concepts that underpin our proposed method, focusing on the Information Bottleneck Theory (IB) and the Reinforce Algorithm, both of which are crucial for understanding the design philosophy of AdaTok.

**Information Bottleneck Theory.** The IB theory suggests that an optimal representation of data should balance between compressing the input information and retaining the relevant information for the task at hand. Specifically, the theory advocates minimizing irrelevant information while preserving key features that contribute to task performance. This trade-off is formalized by the following loss function:

$$\mathcal{L} = \alpha I(X; Z) - I(Y; Z), \tag{1}$$

where $I(X; Z)$ is the mutual information between the input $X$ and the learned representation $Z$, and $I(Y; Z)$ is the mutual information between the output $Y$ and the representation $Z$. The parameter $\alpha$ controls the balance between these two terms. In the context of visual tokenization, this principle helps design a reward function that encourages both compression (minimizing token length) and

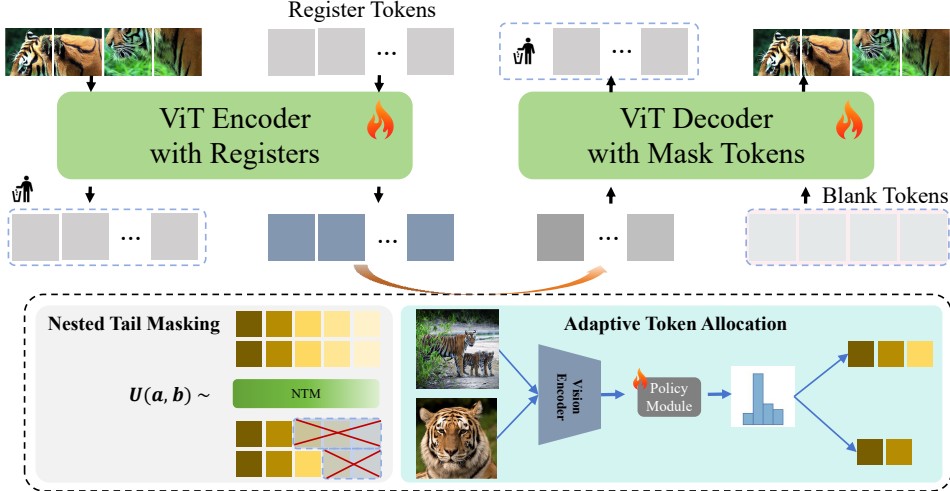

Figure 2: **Framework of AdaTok.** It primarily consists of two modules: Tail Token Masking and Adaptive Token Allocation, which adaptively encode images into flexible tokens. AdaTok learns a flexible representation via Nested Tail Masking with a variable ratio from $a$ to $b$, upon which a Policy Module is trained to autonomously select the optimal token length for each image.

relevance retention (maintaining the quality of the image representation), which is essential for efficient image encoding.

**Reinforce Algorithm.** The Reinforce algorithm is a policy gradient method used to optimize stochastic policies by directly adjusting the parameters of the policy to maximize expected rewards. In this method, a policy $\pi_\theta(a|s)$ defines the probability of taking action $a$ given state $s$, where $\theta$ represents the parameters of the policy. The goal is to maximize the expected cumulative reward, denoted as the return $G_t$, over time. This return is typically the sum of the rewards obtained from time step $t$ onward, possibly discounted by a factor $\gamma$ to prioritize more immediate rewards.

The core of the Reinforce algorithm is to compute the gradient of the expected return with respect to the policy parameters. This is done by evaluating the log-probability of the taken action, weighted by the observed return $G_t$. The gradient of the expected return is expressed as:

$$\nabla_\theta J(\theta) = \mathbb{E}\left[\nabla_\theta \log \pi_\theta(a_t|s_t) \cdot G_t\right], \tag{2}$$

where $\pi_\theta(a_t|s_t)$ is the probability of selecting action $a_t$ at state $s_t$, and $G_t$ is the return starting from time step $t$. By adjusting the policy parameters $\theta$ using gradient ascent, actions that yield higher returns are reinforced, while less favorable actions are discouraged.

In the context of adaptive token allocation, we apply the Reinforce algorithm to optimize the number of tokens used for encoding images. By updating the policy based on the observed rewards—where the reward function combines both compression and quality—this approach enables efficient allocation of tokens for image encoding, adapting the token length dynamically to the content.

## 4 METHOD

Here, we introduce **AdaTok**, a novel flexible representation paradigm with elastic tokens. As shown in Figure 2, the tokenizer module of AdaTok adaptively encodes images into varying lengths while maintaining the quality of each sample, significantly enhancing the efficiency of the representations. This design not only prevents the over-encoding of visual representations but also accelerates the downstream generation process effectively. Leveraging such elastic representations, the VLM model can effectively ensure that the image generation process stops at the appropriate point. Crucially, our elastic visual representation brings the vision modality into greater structural alignment with language. This convergence towards a variable-length format is a critical step in laying the groundwork for more unified and efficient VLMs.

The elastic tokenizer framework integrates two core mechanisms: *Nested Tail Masking* and *Adaptive Token Allocation*. *Nested Tail Masking* is a training strategy that involves randomly masking the

tail tokens, thereby stimulating the model's ability to reconstruct sequences with varying token lengths. On the other hand, *Adaptive Token Allocation* allows the model to autonomously determine the minimal number of tokens required to maintain quality, optimizing its token usage efficiency. We depart from conventional 2D-structured tokenizers where each token represents a local patch, making them indispensable for a complete reconstruction. Instead, we adopt a 1D, structure-agnostic tokenizer like TiTok (Yu et al., 2025; Darcet et al., 2023). Here, each token captures global image information, granting us the flexibility to discard some tokens while still being able to reconstruct the entire image, merely with a reduction in quality.

## 4.1 Nested Tail Masking (NTM)

To enable our model to handle variable-length inputs, a flexible training strategy is required. Prior methods like Tail Token Drop (Rippel et al., 2014; Kusupati et al., 2022) implicitly induce an information hierarchy, but they typically apply a uniform drop strategy across an entire batch. This approach is incompatible with our goal of sample-level adaptation, as it prevents the use of per-sample policies required for our subsequent Adaptive Token Allocation (ATA) module. To overcome this, we propose **Nested Tail Masking (NTM)**, a sample-specific masking strategy that operates on the attention mechanism, which our experiments confirm is the key component for this training effect.

**1D Suffix Token Encoding.** Following the 1D tokenization paradigm (Yu et al., 2025), we represent an image as a sequence of tokens. However, instead of prefixing learnable tokens, we append them as suffixes. Image information is encoded into these learnable suffix tokens via a Vision Transformer (ViT) encoder. The process is formulated as:

$$z = \text{Encoder}(p_1, \ldots, p_n, q_1, \ldots, q_m)[n:] \tag{3}$$

where $\{p_i\}_{i=1}^n$ are the input image patches and $\{q_i\}_{i=1}^m$ are the learnable suffix tokens. The final output tokens $z \in \mathbb{R}^{m \times d}$ constitute the image's representation, while the initial patch embeddings are discarded. We perform this tokenization in a well-structured VAE latent space (Yao et al., 2025) rather than the pixel space to improve efficiency and avoid the information loss associated with vector quantization.

**Randomized Tail Masking.** To foster representational flexibility, we train the model to reconstruct the image from a variable number of its suffix tokens. During each training step, for each sample in the batch, we randomly determine a number of tokens to drop, $k \sim U(k_{\min}, m-1)$. Crucially, we only mask tokens from the *tail* of the sequence. This encourages an information hierarchy where earlier tokens ($q_1, q_2, \ldots$) capture more general, high-importance information, while later tokens encode finer details. The resulting token sequence is:

$$z_{\text{masked}} = (q_1, q_2, \ldots, q_{m-k}, M_{m-k+1}, \ldots, M_m) \tag{4}$$

where $\{M_i\}$ are placeholder mask tokens. This "nested" property—where a representation of length $L$ is always a prefix of a representation of length $L+1$—is essential for our adaptive framework.

**Mask-Aware Training and Loss.** To ensure that the masked tokens do not negatively influence the model, we employ a Key Padding Mask within all self-attention layers of the subsequent decoder. This effectively makes the model "blind" to the masked positions. The model is then trained end-to-end to reconstruct the original image from the unmasked tokens. The overall training objective for this stage, $\mathcal{L}_{NTM}$, is a combination of reconstruction and perceptual losses:

$$\mathcal{L}_{NTM} = \mathcal{L}_{\text{recon}}(x, \hat{x}) + \lambda_p \mathcal{L}_P(x, \hat{x}) + \lambda_g \mathcal{L}_G(\hat{x}) \tag{5}$$

where $x$ and $\hat{x}$ are the input and reconstructed images, respectively. $\mathcal{L}_{\text{recon}}$ is a pixel-level reconstruction loss (e.g., L2), while $\mathcal{L}_P$ and $\mathcal{L}_G$ are the perceptual and adversarial losses, which ensure high-fidelity and realistic outputs. The weights $\lambda_p$ and $\lambda_g$ balance the contribution of each term.

## 4.2 Adaptive Token Allocation (ATA)

After training with Nested Tail Masking (NTM), our model can represent images with varying token lengths. We leverage this capability through the Adaptive Token Allocation (ATA) mechanism, which aims to find the optimal number of tokens for each image. As shown in Figure 3, the relationship between token length and reconstruction quality (e.g., rMSE) exhibits clear diminishing returns, indicating that a sample-specific optimal trade-off exists. To dynamically find this trade-off, we

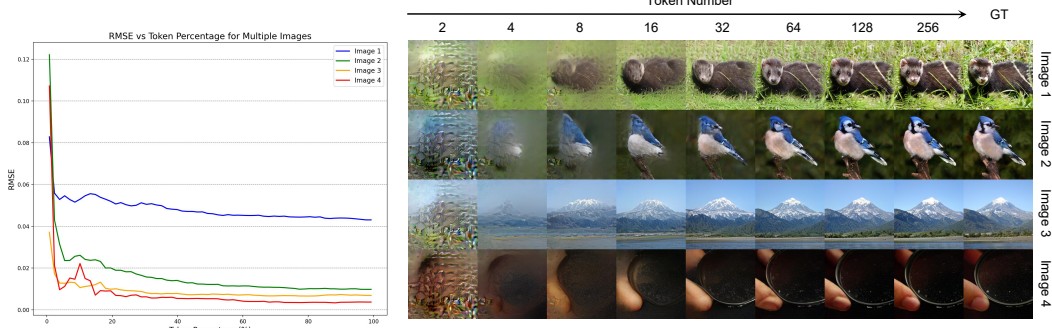

Figure 3: **rMSE *vs.* Token Usage for Variable Samples.** The left shows rMSE curves for different samples, reflecting the relationship between reconstruction quality and token length. The right shows qualitative results, clearly illustrating quality changes.
introduce a **policy network** that learns to select the most appropriate token length for each image based on its unique content.

We frame this dynamic selection process as a reinforcement learning (RL) problem. The policy network $\pi(L|f, \theta)$, conditioned on image features $f$, outputs a probability distribution over possible token lengths $L$. A specific length $L_{keep}$ is then sampled for each image: $L_{keep} \sim \pi(L|f, \theta)$.

**Reward Function Design.** To guide the policy, the design of the reward function $R(L, f)$ is paramount. It must encapsulate the core trade-off between compression (low token length) and fidelity (low reconstruction loss). We explore two distinct designs, each with a different inductive bias.

**1. Continuous Reward Function.** Our first design is a simple yet effective continuous function that linearly penalizes both reconstruction loss and token length. This approach provides a smooth, dense reward signal across the entire action space. It is defined as:

$$R_{\text{continuous}}(L, f) = -rl \cdot \alpha - (tl/L_{\max}) \cdot \beta \tag{6}$$

where $rl$ is the pixel-level reconstruction loss (e.g., MSE), $tl$ is the token length, $L_{\max}$ is the maximum possible length (e.g., 256), and $\alpha, \beta$ are weighting hyper-parameters. This function encourages the policy to continuously seek a state where both losses are jointly minimized. The convergence properties of this function are discussed in Appendix B.2.

**2. Structured Threshold-based Reward.** Our second design introduces a "satisfaction threshold" $rl_{\text{base}}$ (e.g., an MSE of 0.02), which defines a target for "acceptable" reconstruction quality. This transforms the problem from a simple joint minimization into a more structured, two-stage optimization, mimicking a human-like decision process: "first, meet the quality standard; then, optimize for efficiency." The reward is defined as:

$$R_{\text{structured}}(L, f) = \begin{cases} 1 + (L_{\max} - tl)/L_{\max} & \text{if } rl \leq rl_{\text{base}} \\ -(rl - rl_{\text{base}}) \cdot \gamma & \text{if } rl > rl_{\text{base}} \end{cases} \tag{7}$$

When the quality target is met ($rl \leq rl_{\text{base}}$), the agent receives a primary reward of 1, plus an "efficiency bonus" for using fewer tokens. If the target is not met, it is penalized based on how far the quality is from the threshold and, to a lesser extent, by the number of tokens used. To ensure stable training, we implement this piece-wise function using a smooth 'sigmoid'-based approximation to avoid discontinuities at the threshold (see Appendix B.3 for details).

**Policy Optimization with Relative Reward Gain.** Regardless of the chosen reward function $R(L, f)$ (either $R_{\text{continuous}}$ or $R_{\text{structured}}$), a naive application of the REINFORCE algorithm by maximizing the expected absolute reward $\mathbb{E}[R(L, f)]$ is prone to high variance. To stabilize training and encourage more effective sample-level optimization, we use a baseline to compute the *relative reward gain* (or advantage). This measures how much better a specific action (token length $L$) is compared to the policy's average behavior.

Specifically, we define a baseline reward computed using the policy's mean token length, $L_{\text{mean}} = \mathbb{E}_\pi[L]$. The relative gain, which we use as our learning signal, is then:

$$A(L, f) = R(L, f) + \mu(R(L, f) - R(L_{\text{mean}}, f)) \tag{8}$$

Table 1: **Reconstruction and Generation Comparisons across Tokenizers.** This table compares different methods including 1d and 2d tokenizers, metrics contain codebook size, token counts, token dimensions, and reconstruction quality and generation quality (measured by rFID and gFID). If the codebook size is none, the tokenizer is continuous. The best results are **bold**.

| Tokenizer | #Params | Codebook | #Tokens | Elastic | rFID↓ | Generator | gFID↓ |
|---|---|---|---|---|---|---|---|
| 2d tokenizers | | | | | | | |
| VQGAN | 85M | 1024 | 256 | ✗ | 7.94 | LDM-8 | 15.78 |
| MaskGIT | 227M | 1024 | 256 | ✗ | 2.12 | MaskGIT | 6.18 |
| LlamaGen | 72M | 16384 | 256 | ✗ | 2.28 | LlamaGen | 3.06 |
| ImageFolder | 362M | 4096 | 286 | ✗ | 0.80 | VAR-d16 | 2.60 |
| VAR | 310M | 4096 | 680 | ✗ | 0.90 | VAR-d16 | 3.30 |
| CAT | 187M | - | - | ✓ | 0.46 | - | - |
| 1d tokenizers | | | | | | | |
| TiTok-S | 83M | 4096 | 128 | ✗ | 1.71 | MaskGiT | **1.97** |
| TiTok-B | 204M | 4096 | 64 | ✗ | 1.70 | MaskGIT | 2.48 |
| TiTok-L | 641M | 4096 | 32 | ✗ | 2.21 | MaskGIT | 2.77 |
| ALIT | - | 4096 | 256 | ✓ | 8.25 | - | - |
| FlexTok | 341M | 64000 | 32 | ✓ | 4.20 | FlexTok | 3.83 |
| One-D-Piece | 83M | 4096 | 256 | ✓ | 1.48 | MaskGiT | 2.67 |
| Semanticist | - | - | 256 | ✓ | 0.72 | LlamaGen | 2.57 |
| **AdaTok** | 83M | - | 256 | ✓ | **0.42** | DiT | 2.32 |

where $\mu$ is the group reward difference scale with the mean length and $R(L_{\mathrm{mean}}, f)$ is computed on a batch basis and treated as a detached constant during backpropagation. This formulation pushes the policy to not just find a "good" token length, but to find a length for each sample that is demonstrably superior to a one-size-fits-all, average approach.

The policy is optimized using the REINFORCE algorithm with this advantage signal. Furthermore, to encourage exploration, we add an entropy regularization term $H(\pi)$. The final loss function for our ATA module is:

$$\mathcal{L}_{ATA} = \underbrace{-\mathbb{E}_\pi[\log \pi(L|f; \theta) \cdot A(L, f)]}_{\text{Policy Gradient (REINFORCE)}} \underbrace{-\lambda(t) \cdot H(\pi)}_{\text{Entropy Regularization}} \tag{9}$$

where $\lambda(t)$ is a decaying function that controls the exploration-exploitation trade-off. This complete approach enables the model to dynamically and efficiently adjust token lengths on a per-image basis.

## 5 EXPERIMENTS

In this section, we introduce the details of the implementation of AdaTok, and the main results will also be discussed.

**Implementation Details.** We conduct fair training by strictly following the parameter settings described in their respective papers like TiTok (Yu et al., 2025). Specifically, the codebook size is set to 4096, and we employ the AdamW optimizer with an initial learning rate and a weight decay of 4e-5. In addition, a warm-up strategy is applied during the first phase of training, and the batch size of all stages is 256. Besides, in our experiments, $\alpha, \beta, \gamma, \mu$ are set to 20, 0.2 and 10, 0.1 respectively. The tokenizer training is divided into three stages: stage 1 is trained with 500k iterations for TTM, stage 2 is trained for 20k iterations for ATA. Since our method does not need vector quantization, the training cost is much lower than TiTok and One-D-Piece. All experiments are conducted on a hardware setup with 8 Nvidia H800 GPUs.

### 5.1 OVERALL PERFORMANCE

To comprehensively evaluate AdaTok, we benchmark it against a diverse set of baseline methods representing major paradigms in visual tokenization, including seminal 2D models (e.g., VQGAN), recent 1D series (e.g., TiTok), and both discrete and continuous approaches with fixed or elastic lengths (Table 1). As presented, AdaTok demonstrates state-of-the-art performance across the board. In reconstruction, it achieves a record-low Frechet Inception Distance (rFID) of 0.42, significantly

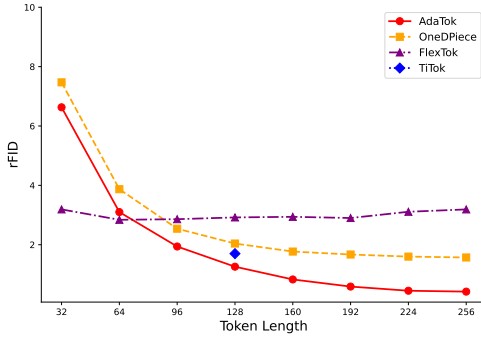

Figure 4: rFID Analysis across different methods with varying token lengths.

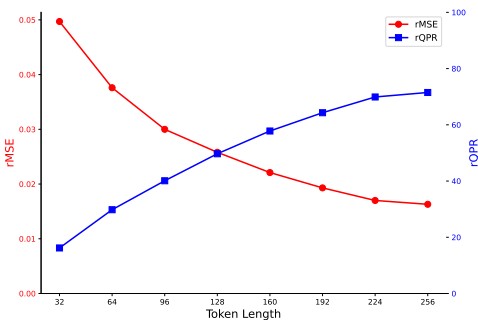

Figure 5: rMSE and rQPR curves across varying token lengths.

outperforming leading methods like CAT (0.46) and MAR (0.53) with an efficient model size of only 83M parameters. Furthermore, these high-quality tokens provide a robust foundation for downstream generative tasks. When used with a DiT-based model, AdaTok achieves an impressive generation FID (gFID) of 2.33 for training 200 epochs, surpassing many specialized elastic baselines such as FlexTok (3.83) and OneDPiece (2.67). These results validate that our adaptive representation strategy excels in both reconstruction fidelity and generative capability, positioning AdaTok at the forefront of visual tokenization.

**Elasticity.** Our method offers two primary advantages. First, it significantly surpasses existing tokenizers, especially 1D structure-free models, in both reconstruction and generation performance. Second, and most critically, it introduces an exceptional level of flexibility by supporting reconstruction from any number of tokens in the 32 to 256 range. We validate this by comparing our model against similarly-sized competitors: TiTok-S, One-D-Piece, and FlexTok-d12-d12, with the results shown in Fig 4. As is evident, our method significantly outperforms other approaches in the 96-256 token range, while also demonstrating strong representational flexibility. It is worth noting that although FlexTok performs competitively in the 32-64 token range, it is a tokenizer with a weaker pixel-level reconstruction objective. Therefore, at such low token counts, a direct comparison using the rFID metric becomes less meaningful. Also, compared with the only elastic continuous tokenizers, CAT, the performance of AdaTok also surpasses it and could select any tokens from 32 to 256.

## 5.2 ABLATION STUDY

| NTM | ATA | Elastic | #Tokens | Reward↑ | RT↓ |
|---|---|---|---|---|---|
| Baseline | | | 256 | 0.726 | **1.8** |
| ✓ | | ✓ | 143 | 0.579 | 4.2 |
| ✓ | +Heur | ✓ | 100 | 0.750 | 70.2 |
| ✓ | ✓ | ✓ | **177** | **0.819** | 4.3 |

Table 2: **Abalation Experiments.** RT means runtime (ms).

To validate the contributions of our key modules, NTM and ATA, we conduct an ablation study summarized in Table 2. The baseline model, using a fixed 256 tokens, is fast (1.8 ms) but inefficient, achieving a reward of 0.726. Introducing NTM provides representation flexibility, but naively applying it (truncating to 143 tokens) causes the reward to drop to 0.579, powerfully demonstrating that an intelligent allocation strategy is crucial to properly leverage this flexibility. While a heuristic-based binary search ('+Heur') can boost the reward to 0.750, its prohibitive runtime of 70.2 ms makes it impractical. In contrast, our full AdaTok model, combining NTM with ATA, achieves the best of all worlds: it secures the highest reward (0.819) while maintaining a fast runtime of just 4.3 ms, which is over 16 times faster than the heuristic approach. This clearly shows that NTM is essential for providing flexibility, and ATA is the critical component for exploiting it efficiently and effectively, striking a superior balance between performance and speed.

## 5.3 ANALYSIS

**Visualizing Progressive Reconstruction.** We investigate the hierarchical nature of the representations learned by NTM through a progressive reconstruction analysis. As visualized in Figure 11 (Appendix), using an incrementally larger set of initial tokens (e.g., 32, 64, 128) reveals a distinct coarse-to-fine behavior: global structures are captured first, followed by finer details. This visual observation

is supported by quantitative metrics. As expected, the reconstruction MSE (rMSE) consistently decreases with token length. To better quantify the notion of "progressive quality improvement," we propose the reconstruction Quality Pass Rate (rQPR): the proportion of images achieving an rMSE lower than a predefined threshold such as 0.02. As illustrated in Fig. 5, the rQPR shows a steady climb as token length increases, demonstrating that a growing fraction of the dataset meets the quality standard. This combined evidence strongly indicates that NTM successfully orders information from most to least critical, which is essential for our subsequent adaptive allocation.

**Analysis of the Reward Landscape and Policy Learning.** To understand the behavior of our Adaptive Token Allocation (ATA) module, we first analyze the reward landscape that guides the policy network. We plot our proposed reward functions, $R_{\text{continuous}}$ and $R_{\text{structured}}$, as a function of token length over the validation set (Fig. 6). Both functions exhibit a clear **concave** shape, initially increasing as more tokens improve reconstruction quality, and then decreasing as the penalty for longer sequences begins to dominate.

This sample-level policy learning proves highly effective. We demonstrate this by comparing the actual reward achieved by AdaTok against a "fixed-length" baseline, where all images are forced to use the same number of tokens—a value equal to the average token length chosen by our policy. In this figure, the reward achieved by our adaptive policy is **consistently higher** than that of the fixed-length baseline, despite both methods operating at the same average token budget. This highlights the core advantage of AdaTok: by dynamically allocating more tokens to complex images and fewer to simple ones, our policy effectively "cherry-picks" the high-reward outcomes for each sample. This results in a superior overall performance that a one-size-fits-all approach cannot achieve, proving the value of dynamic, per-image adaptation.

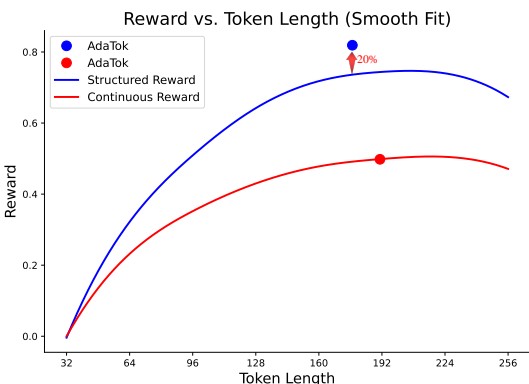

Figure 6: Analysis of the Reward Landscape and Learned Policy Distribution.

## 6 CONCLUSION

In this paper, we present *AdaTok*, a novel framework for adaptive image representation using variable-length tokens. By combining *Nested Tail Masking (NTM)* and *Adaptive Token Allocation (ATA)*, AdaTok enables the model to autonomously determine the optimal token length required to preserve image quality. The key innovation of AdaTok lies in its ability to dynamically adjust token lengths based on the quality of the image, utilizing a reinforcement loss inspired by the Information Bottleneck Theory to balance token count and reconstruction quality. Our experimental results show that AdaTok outperforms existing methods in terms of generation speed and quality, achieving a significant reduction in token usage without compromising reconstruction fidelity. The flexibility of AdaTok makes it a promising approach for improving efficiency in both visual-textual alignment and autoregressive generation tasks. Future research can explore alternative frameworks beyond next-token prediction and extend AdaTok to tasks like video generation, further enhancing its applicability in multimodal systems.

**Limitations.** Our proposed AdaTok further advances the alignment between images and textual forms, potentially serving as a new paradigm within a unified multimodal framework in the future. However, there are still some limitations that need to be addressed: a) Integrating AdaTok's variable-length tokens with standard generative models requires special consideration. For example, autoregressive models need an [EOS] token to halt generation, while non-autoregressive models require careful padding and masking schemes to handle the varied sequence lengths. c) We have not conducted large-scale experiments on text-to-image generation with vision-language models (VLMs); future work could explore whether the model can adaptively allocate image tokens based on the information density of the input text.

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

APPENDIX CONTENTS FOR ADATOK

## A  STATEMENT ON THE USE OF AI TOOLS

A large language model (LLM) was used to aid in the language polishing of this manuscript. Its role was limited to improving grammar, clarity, and readability. All scientific ideas, methodologies, results, and conclusions were conceived and articulated by the human authors, who are solely responsible for the content of this paper.

## B  CONVERGENCE ANALYSIS OF REWARD FUNCTIONS

In this section, we analyze the convergence properties of the two reward functions proposed in the main text. Our goal is to demonstrate that for a given image, both reward functions $R(L)$ with respect to the token length $L$ (denoted as $tl$ for simplicity) are **strictly concave**. A strictly concave function has a unique global maximum. This property is crucial as it ensures that the policy network has a stable and well-defined optimization target, thereby guaranteeing the convergence of the policy to an optimal length.

### B.1  UNDERLYING ASSUMPTION: DIMINISHING RETURNS

Our analysis rests on a fundamental and empirically validated assumption: the relationship between token length ($tl$) and reconstruction loss ($rl$) exhibits **diminishing returns**. This means that as we add more tokens, the reconstruction loss decreases, but at a progressively slower rate. Mathematically, this implies that $rl(tl)$ is a strictly convex and decreasing function of $tl$:

1. $\frac{\partial (rl)}{\partial (tl)} < 0$

2. $\frac{\partial^2 (rl)}{\partial (tl)^2} > 0$

This convexity is intuitively clear: the first few tokens capture the most critical information, leading to a large drop in loss, while later tokens only add minor details, resulting in marginal improvements. This behavior is empirically verified in Figure 3 of the main text.

### B.2  ANALYSIS OF $R_{\text{CONTINUOUS}}$

The continuous reward function is defined as:

$$R_{\text{continuous}}(tl) = -rl(tl) \cdot \alpha - (tl/L_{\max}) \cdot \beta \tag{10}$$

where $\alpha, \beta > 0$ are constants. Its second derivative with respect to $tl$ is:

$$\frac{\partial^2 R_{\text{continuous}}}{\partial (tl)^2} = -\alpha \cdot \frac{\partial^2 (rl)}{\partial (tl)^2} \tag{11}$$

Given our assumption that $rl(tl)$ is strictly convex ($\frac{\partial^2 (rl)}{\partial (tl)^2} > 0$) and $\alpha > 0$, it directly follows that:

$$\frac{\partial^2 R_{\text{continuous}}}{\partial (tl)^2} < 0 \tag{12}$$

This proves that $R_{\text{continuous}}$ is a strictly concave function of the token length $tl$ and thus possesses a unique global maximum.

### B.3  ANALYSIS OF $R_{\text{STRUCTURED}}$

The structured reward function is defined as a piecewise function:

$$R_{\text{structured}}(tl) = \begin{cases} R_{\text{high}}(tl) = 1 + (L_{\max} - tl)/L_{\max} & \text{if } rl(tl) \leq rl_{\text{base}} \\ R_{\text{low}}(tl) = -(rl(tl) - rl_{\text{base}}) \cdot \gamma & \text{if } rl(tl) > rl_{\text{base}} \end{cases} \tag{13}$$

This function is not differentiable at the boundary point where $rl(tl) = rl_{\text{base}}$. However, we can still prove its strict concavity by analyzing each piece and the behavior at the boundary.

Let $tl^*$ be the token length such that $rl(tl^*) = rl_{\text{base}}$. Since $rl(tl)$ is a decreasing function, this boundary point $tl^*$ is unique. The function is then defined over two intervals: $[0, tl^*]$ and $(tl^*, L_{\max}]$.

1. **Analysis of each piece:**

   - For $tl \in (tl^*, L_{\max}]$, the function is $R_{\text{low}}(tl)$. Its second derivative is $\frac{\partial^2 R_{\text{low}}}{\partial (tl)^2} = -\gamma \cdot \frac{\partial^2 (rl)}{\partial (tl)^2}$. Since $\gamma > 0$ and $rl(tl)$ is strictly convex, this piece is **strictly concave**.

   - For $tl \in [0, tl^*]$, the function is $R_{\text{high}}(tl)$. Its second derivative is $\frac{\partial^2 R_{\text{high}}}{\partial (tl)^2} = 0$. This piece is linear, and thus (non-strictly) **concave**.

2. **Behavior at the boundary $tl^*$:** At the boundary point $tl^*$, the function is continuous:

$$\lim_{tl \to tl^{*+}} R_{\text{low}}(tl) = -(rl(tl^*) - rl_{\text{base}}) \cdot \gamma = 0$$
$$R_{\text{high}}(tl^*) = 1 + (L_{\max} - tl^*)/L_{\max}$$

There appears to be a discontinuity if the formula is as written. Assuming the user intended continuity, a formulation such as $R_{\text{low}} = R_{\text{high}}(tl^*) - (rl(tl) - rl_{\text{base}}) \cdot \gamma$ would be continuous. However, even with a discontinuity, the argument about the shape holds. Let's analyze the slopes (first derivatives) at the boundary:

$$\left. \frac{\partial R_{\text{low}}}{\partial (tl)} \right|_{tl \to tl^{*+}} = -\gamma \cdot \left. \frac{\partial (rl)}{\partial (tl)} \right|_{tl \to tl^{*+}}$$
$$\left. \frac{\partial R_{\text{high}}}{\partial (tl)} \right|_{tl \to tl^{*-}} = -1/L_{\max}$$

Since $\frac{\partial (rl)}{\partial (tl)} < 0$ and $\gamma > 0$, the slope of $R_{\text{low}}$ is positive. The slope of $R_{\text{high}}$ is negative. This means the slope decreases as we cross the boundary from left to right, which is a key property of concave functions.

Because the function is composed of a concave piece and a strictly concave piece, and the slope decreases at their joining point, the overall function $R_{\text{structured}}(tl)$ is strictly concave. It may have a "sharp peak" (a kink) at $tl^*$, but it still has a single, well-defined global maximum.

### B.4 CONCLUSION

Both proposed reward functions, under the empirically supported assumption of diminishing returns in reconstruction quality, are shown to be strictly concave with respect to the token length. This mathematical property is crucial, as it guarantees the existence of a unique optimal token length for any given image, providing a stable and reliable target for our reinforcement learning agent to converge upon, even in the case of the non-differentiable structured reward.

### B.5 DOWNSTREAMING PERFORMANCE

**Linear Probing Evaluation.** To evaluate the representation power and semantic ability of our model, we conduct a linear probing experiment following the protocols used in TiTok and MAE (He et al., 2022). A linear classifier is trained on the output of the encoder for a classification task, such as ImageNet-1K, to measure the linear separability of the encoded representations. This evaluates how well the model captures semantic information and transfers it to downstream tasks. Fig. 8 demonstrates our method alongside all models from TiTok and One-D-Piece. Our model outperforms all their versions in the same linear probing settings, and we also incorporate random tail drop in linear probing to enhance robustness. Since Ada-Tok implicitly ranks token information, applying linear probing does not result in significant information loss and even stimulates the model's scalability. Experimental results validate the linear separability of AdaTok features and their excellent transferability to downstream tasks.

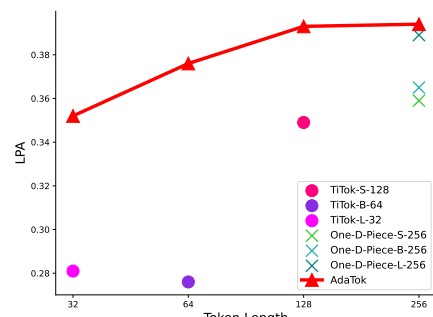

Figure 7: **Class Conditional Generation Results.** Conditions are randomly selected in 1k classes.

**Class-conditional Generation** To validate that our method can indeed enhance downstream tasks, we validate class-conditional generation on ImageNet-1K for comparison. We selected the classic DiT as the generator, and conducted experiments comparing flexible and non-flexible representations using AdaTok. Fig. 7 shows the generated results after training the generator. The model produces high-quality results for many categories, indicating that our flexible representation significantly improves generation efficiency while maintaining quality.

Figure 8: Comparisons of Linear Probing Accuracy (LPA, top-1).

## C    QUALITATIVE RECONSTRUCTION RESULTS

Figure 11 illustrates the reconstruction results of AdaTok across different numbers of tokens, ranging from 2 to 256, with the ground truth (GT) images provided in the rightmost column. The progressive improvement in reconstruction fidelity as the number of tokens increases demonstrates the adaptability and efficiency of AdaTok in capturing finer image details.

At lower token counts (e.g., 2, 4, and 8 tokens), the reconstructions capture only coarse features of the images, resulting in blurry outputs. As the token count increases (16, 32, and beyond), AdaTok begins to recover more structural details, including object shapes and textures. By 256 tokens, the reconstructed images closely approximate the ground truth, showcasing remarkable fidelity and high-quality reconstruction. This progression highlights the ability of AdaTok to flexibly allocate tokens based on the complexity of the content, ensuring efficient and high-fidelity reconstructions across diverse visual inputs.

## D    ENTROPY LOSS ANALYSIS

Figures 9 and 10 demonstrate the effect of progressively decaying entropy loss during training. The entropy loss is carefully designed to promote exploration in the early stages of training, allowing the model to discover diverse tokenization strategies. As training progresses, the entropy loss decays, encouraging the model to converge to a more stable and optimal policy.

Figure 9 shows the gradual reduction in entropy loss over training steps, reflecting the decrease in exploration as the model converges. Correspondingly, Figure 10 illustrates the log probabilities of token selection, which stabilize over time, further validating the model's ability to adaptively learn an efficient tokenization policy. This dynamic adjustment ensures a balance between exploration and exploitation, leading to improved performance across various settings.

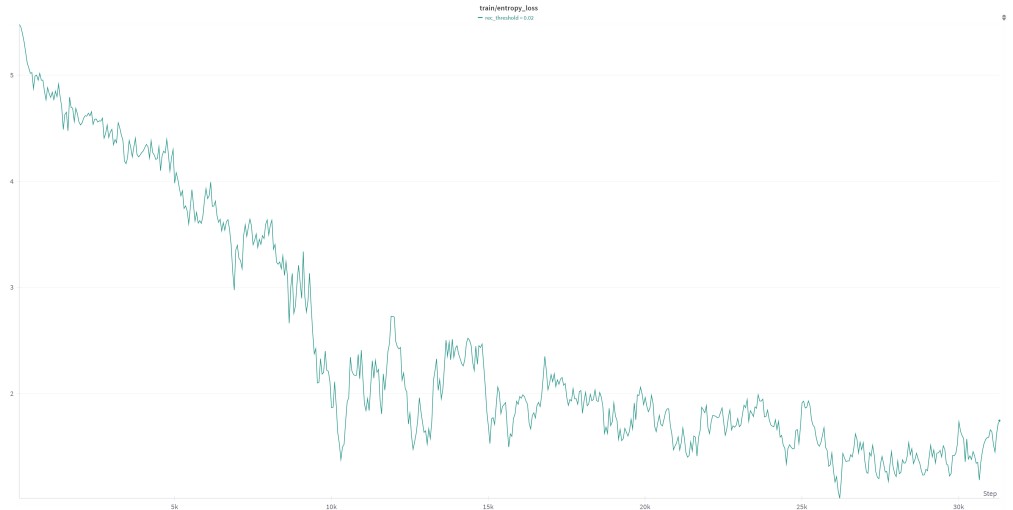

Figure 9: **Entropy loss progression during training.** The gradual decay of entropy loss encourages exploration in early training and promotes convergence in later stages.

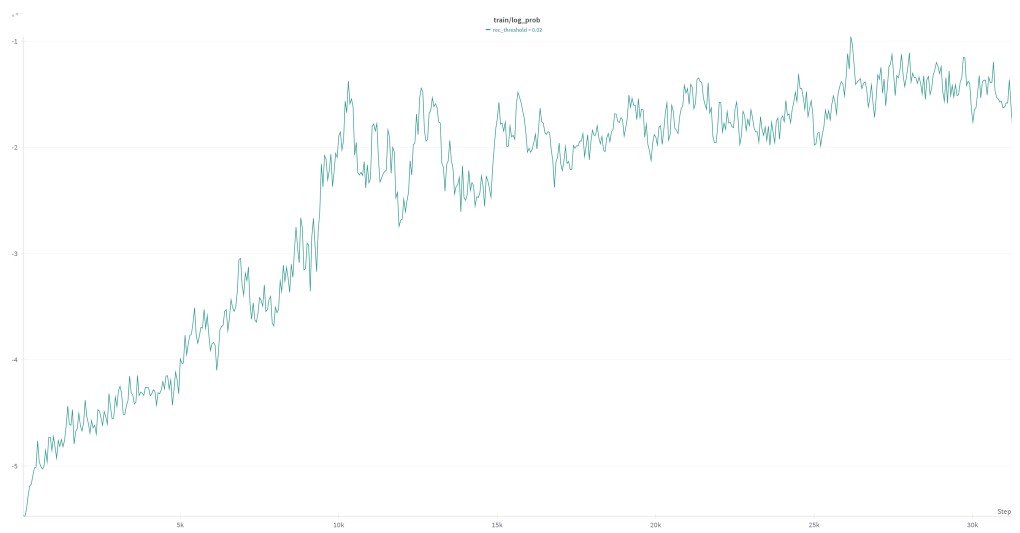

Figure 10: **Log probability evolution.** The stabilization of log probabilities over training steps reflects the model's learned tokenization policy and reduced uncertainty.

## E    DATASET LICENSES

The datasets used for training and evaluating TiTok are summarized below:

**ImageNet-1K**: This dataset spans 1,000 object classes and includes **1,281,167 training images**, **50,000 validation images**, and **100,000 test images**. The training set was utilized for both tokenizer and generator training, while the validation set was employed to compute reconstruction FID scores for tokenizer evaluation. Generation results were further assessed using FID scores, based on pre-computed statistics and scripts from ADM (Dhariwal & Nichol, 2021).

**License**: https://image-net.org/accessagreement

**URL**: https://www.image-net.org/

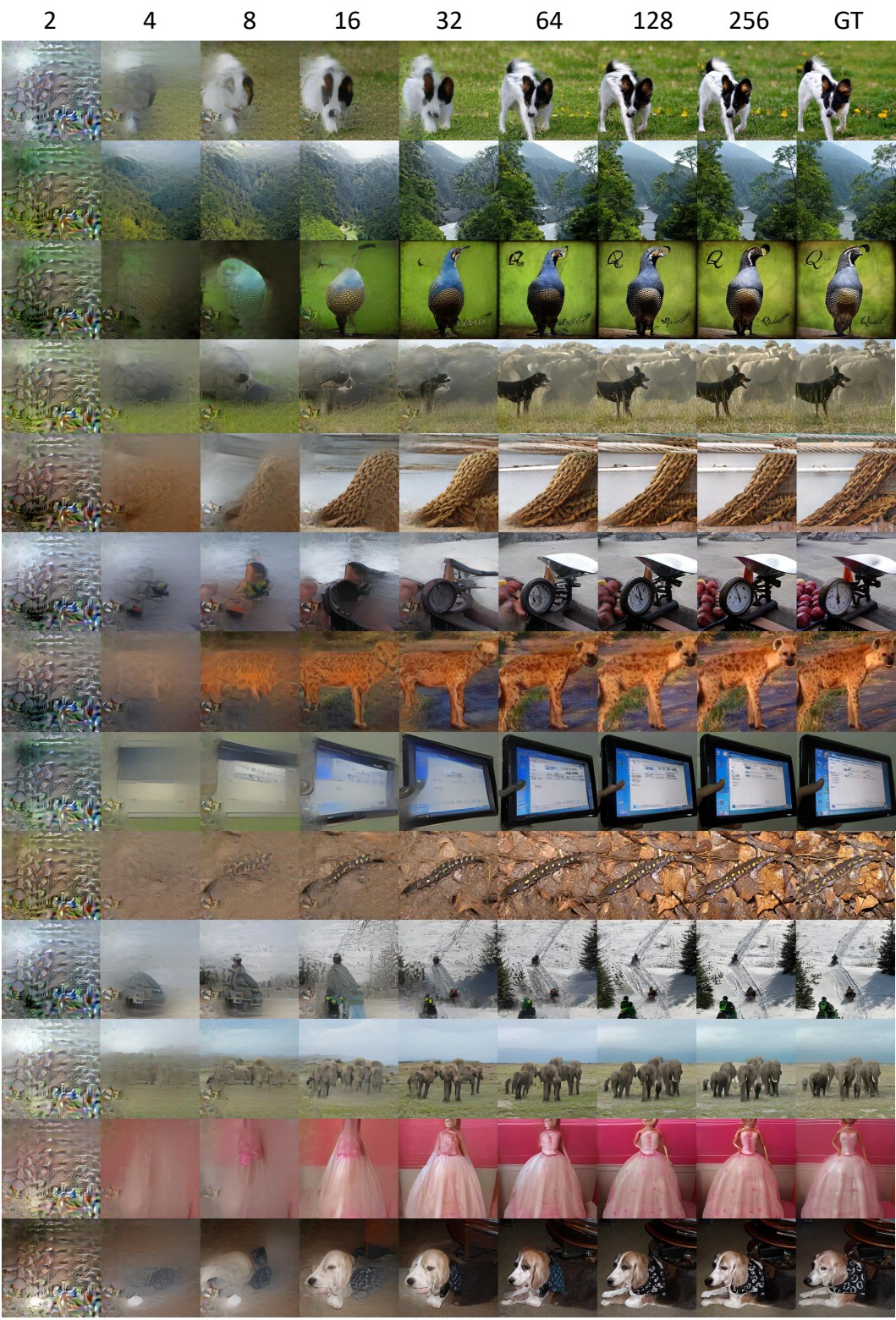

Figure 11: **Reconstruction Results of AdaTok.** The figure demonstrates the progressive reconstruction quality of AdaTok as the number of tokens increases from 2 to 256, with the ground truth (GT) shown in the last column. Higher token counts yield finer details and better fidelity to the original images.

