# OpenReview forum: "ADATOK: ADAPTIVE VISUAL REPRESENTATION WITH QUALITY-PRESERVING DYNAMIC TOKENS"
_ICLR.cc/2026/Conference — ICLR 2026 Conference Withdrawn Submission_

### Official Review · Reviewer_jYkn · 2025-10-22

**Soundness:** 1
**Presentation:** 1
**Contribution:** 2
**Rating:** 2
**Confidence:** 3

**Summary:**

This paper introduces an adaptive size image tokenization approach composed of a variable-length 1D tokenizer similar to One-D-Piece/FlexTok (i.e. TiTok trained with nested token dropout), accompanied by an RL-based representation size selection algorithm that dynamically predicts the number of tokens necessary to achieve acceptable reconstruction fidelity.

**Strengths:**

The motivation for the proposed approach is strong and intuitive: different natural images vary in visual complexity, so the size of their tokenized representation should adapt accordingly. The encoding length selection approach, based on reinforcement learning to learn an input-dependent length which balances a reconstruction quality target and a length penalty, is simple and sound. Furthermore, the ability to predict the target length in a single forward pass of a small policy network is compelling, compared to e.g. a search-based strategy requiring multiple executions of the tokenizer.

**Weaknesses:**

An analysis of the reconstruction quality when using the proposed variable length selection algorithm appears to be completely missing. Since automatic variable length selection is the key contribution of this paper, I consider this to be a critical omission. Specifically, I expect reconstruction quality metrics using the truncated encodings – truncated to the encoding length predicted by your “adaptive token allocation” (ATA) algorithm – to be reported, but I could not find any such results in the paper. *(This is the main reason for low scores in soundness and presentation.)*

To the best of my understanding, the variable-length 1D tokenizer itself appears to be almost identical to previous variable-length 1D tokenizers like One-D-Piece or FlexTok (both cited in the paper), so I do not consider this component to be a significant contribution. Furthermore, key parameters such as the token dimensionality are not reported, which makes evaluation of reconstruction and generation in the context of other tokenizers difficult.

**Questions:**

**Q1.** What is the difference between your tokenizer and One-D-Piece? Is it just the lack of vector quantization?

**Q2.** What is the dimensionality of each token? I could not find this important parameter anywhere in the paper; it is crucial in order to be able to contextualize both the reconstruction and generation metrics with other approaches.

**Q3.** Do you apply any VAE-style KL-regularization to the tokens?

**Q4.** Do you apply causal masking among the 1D tokens?

**Q5.** In L364 you mention a codebook size of 4096, but in my understanding, your tokenizer produces continuous tokens. What does this codebook refer to?

**Q6.** Qualitative reconstruction examples (e.g. figures 1, 3) look visually much worse than results from TiTok-L-32 (or One-D-Piece), yet the rFID reported for your method (0.42) is much better than TiTok-L-32 (2.21). It is possible that my subjective judgement is wrong. However, is there a chance these qualitative figures were produced with a different model/checkpoint than the one used for results in Table 1, or is it possible that there is a problem with your rFID computation? Could you also provide IS or other alternative metrics?

**Q7.** Can you expand on the search-based token allocation? How is it possible that the learned ATA policy can outperform the search over all possible lengths? It should be possible to implement a binary search that finds the minimum number of tokens required to achieve a certain MSE threshold, which should match your reward (2) and therefore match or outperform the learned policy, under the assumption that the MSE decreases monotonically as a function of number of tokens (although this assumption is possibly slightly incorrect due to perceptual and adversarial losses).

**Q8.** Ablation study questions: How was the number “143” chosen? Why does your search approach end up using an average of 100 tokens, which is much lower than the learned policy?

**Q9.** Can you provide details on the DiT generation process, e.g. number of sampling steps, throughput/runtime, and size of generator network?

**C1.** Comment regarding weaknesses section: I appreciate the inclusion of the ablations as well as the reward analysis from Figure 6. However, it is crucial to include at least:
* rFID for reconstructions after truncating the token sequence to the ATA-predicted length
* rFID when truncating uniformly to the mean length predicted by ATA
* Statistics of token sequence length when truncating using ATA (e.g. mean/min/stddev or a histogram)
* The ablation study in 5.2 / Table 2 should include an rFID or MSE column

It would also be appreciated to additionally include qualitative results showing validation images which ATA allocates the smallest number of tokens to (hopefully these should correspond to visually “simple” images) as well as the maximum number of tokens (“complicated” images).

**C2.** Comment regarding “information bottleneck theory”: The connection between IB theory and NTM/ATA was not clear to me. I am not familiar with IB theory, but in my understanding there is no apparent connection between IB theory / eq. (1) and the rest of the paper (beyond superficial similarities/“inspiration”). This makes the IB theory paragraph in the preliminaries section as well as references to IB theory throughout the paper seem unnecessary or misleading. Moreover, IB theory papers are not cited; if you decide to elaborate on the connection to IB theory, you should consider citing the papers by Tishby et al. (e.g. [1]).

**C3.** At least some parts of the appendix appear to be LLM-generated based on the phrase "Assuming **the user** intended..." (emphasis mine) in L665-L666. This content was not reviewed carefully enough by the authors and is therefore of dubious value. It also brings into question the trustworthiness of the rest of the paper.

---

Minor comments:
* The anonymous GitHub link did not work for me (got “The repository is not found”)
* For the structured threshold-based reward, I did not understand why, “if the target is not met, it is penalized [...] to a lesser extent, by the number of tokens used”. I see no dependence on $tl$ in the $rl \geq rl_\mathrm{base}$ case in eq. (7)
* I don’t think the nested dropout from Rippel et al. (2014) is “uniform [...] across an entire batch” (L228). As such, I do not really see a difference between the proposed Randomized Tail Masking and Rippel’s nested dropout, and it seems misleading to claim otherwise.
* Typo: Abalation -> Ablation (in multiple places)
* L245: should it read “prefix” tokens rather than “suffix”?
* L284: missing space between figure caption and main text

---

[1] Naftali Tishby, Noga Zaslavsky. "Deep Learning and the Information Bottleneck Principle." 2015 IEEE Information Theory Workshop. https://arxiv.org/abs/1503.02406

---

### Official Review · Reviewer_E7ca · 2025-10-26

**Soundness:** 2
**Presentation:** 1
**Contribution:** 2
**Rating:** 4
**Confidence:** 5

**Summary:**

This paper proposes AdaTok, a interesting adaptive visual tokenizer that dynamically adjusts the number of tokens per image while maintaining reconstruction quality. It introduces two key modules: Nested Tail Masking (NTM) — trains a one-dimensional image tokenizer to reconstruct from variable-length sequences; Adaptive Token Allocation (ATA) — employs a policy network optimized via REINFORCE and an information bottleneck–inspired reward to predict the optimal token count for each image. Experiments show some improvements over prior 1D tokenizers, with only 83M parameters, while preserving flexibility and speed.

**Strengths:**

1. The author proposes an NTM-based training approach to learn the distribution of token lengths and employs a policy network to control the selection of specific token numbers, which is an interesting idea.

2. The reward construction implicitly maintains the property of information bottleneck theory.

**Weaknesses:**

1. The writing quality is weak. Specifically, although the author claims that the reinforcement learning approach is inspired by the information bottleneck (IB) theory and emphasizes this connection in the abstract and preliminary sections, the reinforcement section lacks both theoretical justification and empirical evidence to support this alignment.

2. From my perspective, the proposed token compression operates in the continuous latent feature space, whereas all baseline methods rely on discrete tokenizers, making the comparison less appropriate.

3. The two stage training is quite complex. Can the model jointly trained with NTM and reinforcement learning?

4. The related work section is insufficient. Important prior studies such as OmniTokenizer (2D) [1] and SweetTok (1D) [2] are not discussed.

[1] Omnitokenizer: A joint image-video tokenizer for visua generation. NIPS 2024.

[2] SweetTok: Semantic-Aware Spatial-Temporal Tokenizer for Compact Video Discretization. ICCV 2025.

**Questions:**

1. Could the REINFORCE module be replaced with a differentiable relaxation (e.g., Gumbel-Softmax)? What is the performance difference?

---

### Official Review · Reviewer_xrQC · 2025-10-30

**Soundness:** 2
**Presentation:** 3
**Contribution:** 2
**Rating:** 4
**Confidence:** 4

**Summary:**

This paper points out that using a fixed number of tokens is inherently suboptimal, as it causes computational redundancy for simple images and risks information loss for complex ones. To address this issue, the paper proposes a new tokenizer, AdaTok, which automatically selects the appropriate number of tokens to represent each image. Experimental results show that the tokenizer achieves strong performance even when the number of tokens is fewer than 256.

**Strengths:**

- The paper is well-written.
 - The tokenzier use a reward function to select dynamic number of the tokens to represnt an image.

**Weaknesses:**

- Since the conclusion in Figure 6 indicates that images represented with fewer than 256 tokens achieve better performance, it is recommended that the authors compare the reconstruction and generation performance of the model using 256 tokens in Table 1.

 - The comparison in Table 1 is unfair and insufficient. As AdaTok is a continuous tokenizer, it should be compared with other continuous tokenizers rather than discrete ones. Moreover, since AdaTok uses DiT as its generative model, the baselines should also adopt DiT for a fair comparison.

 - The conclusion drawn from Figure 4 appears to conflict with the motivation of AdaTok. AdaTok is designed to remove redundant tokens when many tokens are used, yet Figure 4 still shows that performance improves as the number of tokens increases.

 - Since AdaTok can reconstruct images using fewer tokens, it would be valuable to report whether using fewer tokens also reduces inference time and GPU memory consumption.

 - As a new architecture, it is recommended that the authors explore the generative performance of models with different parameter scales to evaluate the scaling capability of AdaTok.

**Questions:**

NA

---

### Official Review · Reviewer_LdmZ · 2025-10-31

**Soundness:** 2
**Presentation:** 3
**Contribution:** 2
**Rating:** 4
**Confidence:** 4

**Summary:**

The paper proposes an approach to variable-length image tokenization, where each image is assigned a different number of representation tokens. Simpler images can be encoded with fewer tokens, while more complex images require more.

The overall approach to adaptive tokenization is conceptually similar to Matryoshka-based (tail-dropping) methods, which learn a maximum-length representation for each image (e.g., 256 tokens) and allow a truncated subset of those tokens to serve as a lossy compressed representation.

The method is evaluated through both reconstruction and generation experiments.

**Strengths:**

The paper is well written overall and addresses an interesting problem: adaptive tokenization.

**Weaknesses:**

Lack of novelty: The paper presents limited conceptual novelty. Adaptive tokenization through tail-dropping or Matryoshka-style approaches has already been explored in prior work, including ElasticTok, One-D-Piece, and FlexTok. Additionally, methods such as ALIT introduce variable token selection at inference time, and recent work like KARL [1] (NeurIPS) performs one-shot token selection based on the desired reconstruction quality. The main new contribution here is the use of an RL policy that jointly optimizes reconstruction loss and a minimum-token regularizer. However, similar RL-based adaptive token selection has been extensively explored in LLM literature (e.g., L1 [2]), and doesn’t contribute to significant novelty.

Figure 1 is ambiguous. It is unclear whether the displayed images are generations or reconstructions. If they are reconstructions, it would be helpful to also show the corresponding original input images. Additionally, the max-token reconstructions appear worse than those using fewer tokens—this is counterintuitive and requires explanation.

The section on information bottleneck theory appears to simply restate rate–distortion theory, which applies generically to both 1D and 2D tokenizers, regardless of whether they use fixed or variable-length representations.

Figure 4: The rFID curve of FlexTok increases as token length increases, which contradicts the trend reported in the FlexTok paper. Clarification would be great.

Prior adaptive tokenization methods either (1) search at inference time for the minimum number of tokens (ElasticTok, FlexTok, One-D-Piece, ALIT), or (2) perform one-shot token selection based on a target reconstruction quality (KARL [1]).In contrast, this paper trains a policy that jointly optimizes reconstruction loss and token count. However, the trade-off between these objectives is controlled by manually chosen hyperparameters (α and β in Eq. (6)), making the approach more restrictive than both prior categories.

Similar to [1], the authors could include an experiment demonstrating how many tokens each method uses to achieve a reconstruction error below a given threshold (this directly supports Section 4.2 (Point 2)).

Missing gFID comparisons at different token counts. The paper reports gFID inconsistently and does not include values across multiple token lengths, limiting comparability with prior work.

FID alone could be insufficient. FID is not always reliable—for example, a rotated version of an image can still yield a good FID score. To allow reviewers to properly assess reconstruction and perceptual quality, the paper should include more comprehensive metrics such as L1, LPIPS, PSNR, SSIM, and/or DreamSim, in addition to FID.

Minor comments:

The following statement requires further justification of why this is true or intuitive: “The need for adaptive tokenization is magnified in modern Vision-Language Models (VLMs), which suffer from a structural mismatch between variable-length text and fixed-length visual inputs.” The authors should explain or cite reference explaining why this mismatch poses a problem and how adaptive tokenization specifically mitigates it.

Please add vertical spacing between lines 284 and 285 (i.e., after the caption for Figure 3) to improve readability.

**Questions:**

- Could you clarify Figure 1?

- Could you discuss the conceptual contribution of the paper in comparison with [1] and [2]?

- Linear Probing evaluation seems incorrect. One-D-Piece reported unexpectedly low linear probing (LP) performance because they did not follow the correct LP evaluation procedure (as acknowledged in their paper, noting a discrepancy with TiTok results). The authors here might have the same issue, which could explain the low LP scores reported. Please, if you have not done it already, could you check the FlexTok supplementary materials for the correct LP protocol? When evaluated properly, ALIT, TiTok, and FlexTok all achieve significantly higher LP performance than what is shown in this paper. Do you think this could explain the results?

---

### Note · Authors · 2025-12-02

**Comment:**

Thanks for the reviewers' comments, and we decide to withdraw and polish it continuely.

**Withdrawal Confirmation:**

I have read and agree with the venue's withdrawal policy on behalf of myself and my co-authors.